# Evaluation of a novel biplex rapid diagnostic test for antibody responses to *Loa loa* and *Onchocerca volvulus* infections

Jérémy T. Campillo[1]*, Marco A. Biamonte[2], Marlhand C. Hemilembolo[3], François Missamou[3], Michel Boussinesq[1], Sébastien D. S. Pion[1], Cédric B. Chesnais[1]

1 UMI 233 TransVIHMI, Université Montpellier, Institut de Recherche pour le Développement (IRD), INSERM Unité 1175, Montpellier, France, 2 Drugs & Diagnostics for Tropical Diseases, San Diego, California, United States of America, 3 Programme National de Lutte contre l'Onchocercose, Direction de l'Épidémiologie et de la Lutte contre la Maladie, Ministère de la Santé et de la Population, Brazzaville, Republic of the Congo

* jeremy.campillo@ird.fr

**Data Availability Statement:** The data and related documentations that support the findings of this study are openly available in DataSuds repository

## Abstract

### Background

Endemic to Central Africa, loiasis, caused by the vector-borne worm *Loa loa*, affects approximately 10 million individuals. Clinical manifestations include transient angioedema (Calabar swellings), migration of the adult worm under the eye conjunctiva (eye worm) and less specific general symptoms. Loiasis presents a significant public health challenge because *L. loa*-infected individuals can develop serious adverse events after taking ivermectin, the drug used to combat onchocerciasis. In this context, alternative interventions and rigorous diagnostic approaches are needed. Diagnosing loiasis is challenging because its main clinical manifestations are sporadic and non-specific. The definitive diagnosis relies on identifying adult worms migrating beneath the conjunctiva, or microfilariae (pre-larvae) in blood smears. However, "occult loiasis" (infection without blood microfilariae) is frequent. Serological rapid antibody diagnostic tests (ARTs) can provide an alternative diagnostic method. We compared a novel ART simultaneously targeting onchocerciasis (IgG4 to Ov-16 and OvOC3261, test line 1) and loiasis (IgG4 to L1-SXP-1, test line 2), called IgG4-SXP-1 biplex test) to the already established *Loa*-ART (all IgG isotypes to Ll-SXP-1, called pan-IgG-SXP-1 test).

### Methodology

Blood samples underwent both ARTs, read qualitatively and semi-quantitatively. Additionally, blood smears, skin snips, Kato-Katz method for soil-transmitted helminthiases identification and eosinophilia measurements were performed. Questionnaires gathered demographic details and loiasis-related signs. ARTs performance was compared using specific loiasis-related signs and microfilaremia as references. Discordances between the two ARTs were investigated using logistic regression models.

(IRD, France) at https://doi.org/10.23708/1WABLS. Data reuse is granted under CC-BY license.

**Funding:** This study was fully supported by the European Research Council (ERC) under the European Union's Horizon 2020 research and innovation program (grant agreement No 949963, grant recipient: CBC). The funders had no role in the study design, data collection, data analysis, data interpretation, or writing of the report.

**Competing interests:** The authors have declared that no competing interests exist.

## Principal findings

Out of 971 participants, 35.4% had *L. loa* microfilaremia, 71.9% had already experienced loiasis-related signs, 85.1% were positive in the pan-IgG-SXP-1 test and 79.4% were positive in the IgG4-SXP-1 biplex test. In the microfilariae-positive population, the sensitivity of the rapid tests was 87.4% for the pan-IgG-SXP-1 test and 88.6% for the prototype IgG4-SXP-1 biplex test. Sensitivity was similar for both ARTs when using eye worm or Calabar swelling as references, but diagnostic performance varied based on microfilaremia levels and occult loiasis. Overall, IgG4-SXP-1 biplex test demonstrated a sensitivity of 84.1% and specificity of 47.6% for loiasis compared to the pan-IgG-SXP-1 test, leading to a Kappa coefficient estimated at 0.27 ± 0.03 for the qualitative results of the 2 ARTs. In the group that tested positive with the Pan-IgG test but negative with the IgG4-specific test, there was a lower prevalence of STH infection (p = 0.008) and elevated eosinophilia (p<0.001) compared to the general tested population.

## Conclusion/Significance

The sensitivity of each test was good (84–85%) but the diagnostic agreement between the two ARTs was poor, suggesting that IgG and IgG4 antibody responses should be interpreted differently. The assessment of the innovative rapid diagnostic IgG4-SXP-1 biplex test, designed for onchocerciasis and loiasis, shows encouraging sensitivity but underlines the necessity for further in vitro assessment.

### Author summary

Loiasis, a disease caused by the parasite *Loa loa* impacting approximately 10 million people in Central Africa, causes transient angioedemas called Calabar swellings and eye worm episodes. Treatment is challenging, particularly in regions where onchocerciasis, another type of filariasis, is also prevalent. We tested a new kind of test that can detect both diseases at once and compared its performance with a previously available test for loiasis. We took blood samples from 971 people living in an area of Congo where loiasis is endemic. Out of the participants, 35.4% had *L. loa* pre-larvae in the blood–known as microfilariae, and 72.0% had experienced loiasis-related signs. The new test demonstrated promise in detecting the disease, albeit with some likelihood of false positives. Additionally, its performance varied according to the density of microfilariae in the blood. While the results exceeded expectations, further testing is essential to ensure its reliability. If validated, this test could prove instrumental in diagnosing both loiasis and onchocerciasis, offering a valuable tool for public health interventions in affected regions.

## Introduction

Loiasis, caused by the parasitic vector-borne nematode *Loa loa*, is a disease exclusively endemic to Central Africa, with over 14 million people residing in high-risk regions [1]. After insemination, adult female *L. loa* worms produce larvae called microfilariae (mfs), which circulate in the bloodstream. In highly endemic communities, more than 30% of the population harbor mfs in their blood. The main clinical signs of loiasis consist of "Calabar swellings",

which are transient angioedemas, and the migration of adult worms under the eye conjunctiva, often referred to as "eye worm" [2,3]. Additionally, there are more general signs such as itching, skin rashes, muscle discomfort, and joint pain. Moreover, loiasis infection has been associated with complications affecting various organs (heart, central nervous system, spleen, and kidneys) and with excess mortality [4–6].

Diagnosing loiasis poses challenges, as clinical signs can be transient and/or non-specific. Definitive diagnosis relies on the morphological identification of adult worms collected during eye worm episodes, or that of mfs in blood smears. However, mfs are not present in the blood during the pre-patent period (6–12 months after the first infective bite by the vector), and may be undetectable during night time due to their diurnal periodicity. Furthermore, about 40% of infected individuals present a so-called "occult loiasis", i.e. despite being infected with sexually reproductive adult worms, they don't show mfs in the peripheral blood, due to a genetic predisposition [7,8].

Loiasis constitutes a significant public health challenge due to its geographical overlap with onchocerciasis and lymphatic filariasis. Efforts to combat these diseases involve mass drug administration (MDA) programs with ivermectin. However, in areas co-endemic with *L. loa*, the use of ivermectin can induce serious neurological adverse events in subjects having a high *L. loa* microfilaremia [9]. While an alternative treatment strategy exists for lymphatic filariasis in co-endemic regions (semi-annual MDA with albendazole alone), onchocerciasis elimination relies only on ivermectin. This poses ethical and logistical challenges to onchocerciasis elimination programs. Current guidelines permit the use of ivermectin-based MDA in areas where *Loa loa* and onchocerciasis coexist, provided that onchocerciasis is meso- or hyperendemic. However, it is essential to establish close surveillance to promptly identify and manage any adverse events following ivermectin administration. Alternatively, a more rigorous Test and not Treat (TaNT) intervention strategy may be used, consisting in a systematic individual screening for *Loa* and/or onchocerciasis before giving treatment [10]. Point-of-care tests are essential for applying such a strategy. Serological rapid antibody diagnostic tests (ARTs) utilizing lateral flow assay technologies are already available for assessing past exposure to *Onchocerca volvulus* or *L. loa*. The onchocerciasis ART detects IgG4 antibodies to the *O. volvulus*-specific antigen Ov-16 [11]. The loiasis ART detects all IgG isotypes (and probably IgM) specific to *Ll*-SXP-1, a validated marker of exposure to *L. loa* [12]. SXP-1 is expressed by infective L3 larvae and induces an immune response in the human host even if the L3 larvae do not mature into adults. The test was formatted as a double-antigen test, with reporter nanoparticles conjugated to SXP-1 and with SXP-1 immobilized at the test line. The test relies on the bidentate ("Y-shaped") nature of antibodies to form a sandwich between nanoparticles and test line and to produce a visual signal; it is therefore not isotype specific. The test was designed to be highly sensitive when read with the naked eye. Additionally, it can be quantified using a smartphone-based chromatographic test reader or a scorecard. This allows users to set a threshold for test line intensity, determining when the test is considered positive. By adjusting this threshold, users can balance sensitivity and specificity according to their needs. SXP-1 has homologs in the other filariae, and when read with the naked eye the rapid test has a specificity of 82–87% compared to *O. volvulus*, *Wuchereria bancrofti*, and *Mansonella perstans* and 100% versus *Strongyloides stercoralis* and endemic and non-endemic negative controls [12]. It has been suggested that an IgG4-specific SXP-1 test could be more specific. While this remains to be verified experimentally, a prototype IgG4 test has been developed. In fact, the prototype was devised as a biplex ART, in which the first test line targets onchocerciasis (IgG4 antibodies to Ov-16 and OvOC3261) and the second test line targets loiasis (IgG4 to L1-SXP-1). We conducted in a large population the first evaluation of this prototype biplex ART and compared the results with the pre-existing loiasis ART.

## Methods

### Ethics statement

The MorLo (Morbidity due to Loiasis) project is an international collaborative study aimed at assessing the prevalence and incidence of *Loa*-related organ-specific complications in rural African areas where loiasis is endemic. This study has been approved by the Ethics Committee of the Congolese Foundation for Medical Research (N˚ 036/CIE/FCRM/2022) and by the Congolese Ministry of Health and Population (N˚ 376/MSP/CAB/UCPP-21). All participants received clear and appropriate information and signed an informed consent form for the study.

### Study area and population

In 2022, a cohort including 991 individuals living in 21 villages located in a radius of 50 kilometers around Sibiti, the capital town of the Lékoumou Division of the Republic of Congo, was initiated. This region was selected because no MDA with ivermectin for onchocerciasis or lymphatic filariasis had ever been implemented. It is endemic for loiasis, hypoendemic for onchocerciasis, non-endemic for schistosomiasis and deworming campaigns are regularly carried out for children to control soil-transmitted helminthiases (STH). Participants had been previously examined for *L. loa* microfilaremia in 2019 during screening surveys for a clinical trial [13]. Individuals with more than 500 *L. loa* mfs per mL of blood in 2019 were matched on sex and age (± 5 years) with two individuals living in the same village identified as amicrofilaremic in 2019.

### Laboratory procedures

A pan-IgG-SXP-1 test (Drugs & Diagnostics for Tropical Diseases, San Diego, California) and an IgG4-SXP-1 + IgG4 to Ov-16 and OvOC3261 biplex test (Drugs & Diagnostics for Tropical Diseases, San Diego, California) were performed for each patient using blood collected in a heparinized tube by antecubital venipuncture (Fig 1). Both tests were read by a single technician using both a qualitative scale (positive or negative) and a semi-quantitative score (0: negative, 1: control line darker than the test line, 2: darkness is even, and 3: test line darker than the control line). In this manuscript, the pan-IgG-SXP-1 test will be referred to as pan-IgG test and the biplex test as IgG4 test.

Blood (50 μL) was also collected by finger prick from each participant and spread on a microscope slide to prepare thick blood smear (TBS) between 10 am and 4 pm to account for the *L. loa* microfilaremia diurnal periodicity. The slides were dried at room temperature, dehemoglobinized and stained with Giemsa stain within 4 hours. All TBS were examined using a microscope at 100× magnification by experienced technicians to count the *L. loa* mfs. Each TBS was read twice and the arithmetic mean of the counts was used for the statistical analysis. Slides with an MFD difference exceeding 30% between the 2 readings were reread blind to the first result. For each patient with a positive onchocerciasis result on the IgG4 test, two skin snips, one at each iliac crest, were collected using a 2 mm Holth-type corneoscleral punch and incubated in saline at room temperature for 24 hours. Emerged mfs were counted using a microscope, and the individuals' microfilarial density (MFD), expressed as mfs per skin snip (mf/ss), was calculated as the arithmetic mean of the two counts.

Finally, for all participants, we measured eosinophilia from blood collected in an EDTA tube using the HemoCue WBC DIFF System (WBC Diff, HemoCue France, Serris, France). STH infections were identified through the microscopic examination of stool specimens using the Kato-Katz method.

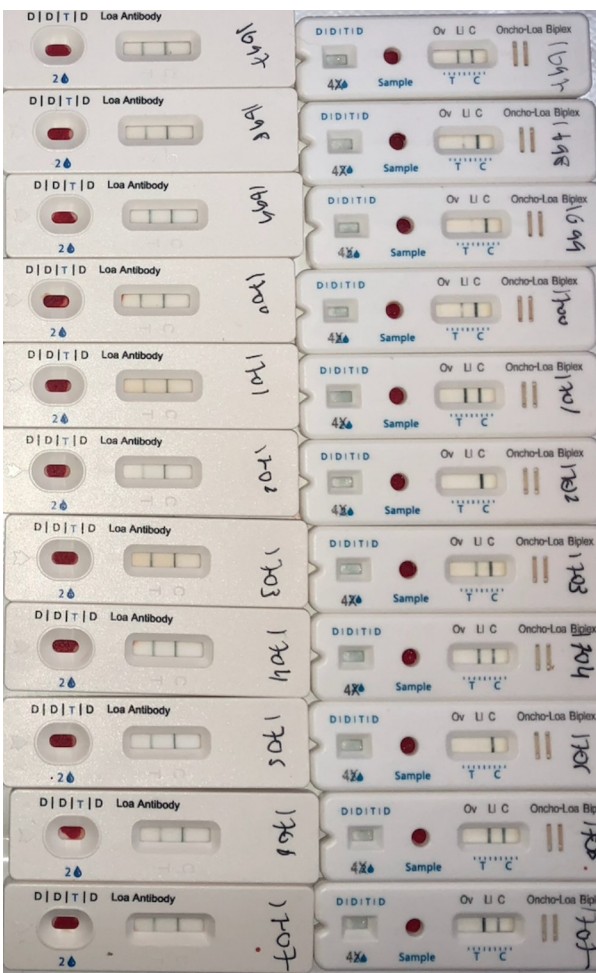

**Fig 1. Example of pan-IgG test and Ig4 test results.** The photo was taken about 30 minutes after the tests were carried out. The tests on the left are the pan-Ig4 test (control strip on the right). The IgG4 test is on the right and, in order from left to right, the strips represent onchocerciasis, loiasis and control.

## Data collection

All participants filled a questionnaire with the assistance of an investigator of the MorLo project. Collected data encompassed demographic information and specific loiasis-related signs: eye worm and transient edema, using the RAPLOA questionnaire [1]. Comprehensive data was gathered regarding the occurrence frequency of these signs both within the past year and throughout the lifetime of each participant.

## Statistical analysis

Qualitative and semi-quantitative results of the pan-IgG test and IgG4 test were compared. Measurements of concordance (inter-rater agreements and Cohen's kappa coefficient) between the two ARTs were calculated with the results presented as binary (positive or negative) and semi-quantitative (0, 1, 2 or 3) categories. Discordances between loiasis results of the pan-IgG test and IgG4 test (positive pan-IgG test and negative IgG4 test, and negative pan-IgG test and positive IgG4 test) were investigated using logistic regression models. Variables included in these models were sex, age (as continuous), *Loa* MFD (as continuous), loiasis-

related signs (history of eyeworm and Calabar swellings in a lifetime), eosinophilia counts (as continuous), STH infections (presence or absence of eggs of any STH) and semi-quantitative result (1 vs. 2 or 3 for convergence issues) of the ART (pan-IgG test or IgG4 test, depending on the nature of the discordance). To evaluate potential cross-reactivity between onchocerciasis and loiasis results on the IgG4 test, we compared the loiasis results obtained from IgG4 test and pan-IgG test among individuals who tested positive for onchocerciasis on the IgG4 test.

Diagnostic performance of the IgG4 test to detect *L. loa* infection has been assessed using the pan-IgG test as the reference. Finally, both ARTs diagnostic performance was analyzed according to specific loiasis-related signs: eye worm, Calabar swellings, and microfilaremia. The different modalities used were: (i) history of eye worm in the past year, (ii) history of eye-worm at least once in the lifetime, (iii) history of Calabar swelling in the past year, (iv) history of Calabar swelling at least once in the lifetime, (v) history of eye worm and/or Calabar swelling at least once in the lifetime, (vi) history of eye worm and/or Calabar swelling at least once in the lifetime and/or presence of microfilaremia (defined as "any sign of *Loa loa* presence"), and (vii) history of eye worm and/or Calabar swelling at least once in the lifetime and absence of microfilaremia (defined as "occult loiasis").

## Results

The two ARTs, blood smear examination, and eosinophilia testing were performed for 971 individuals out of the 991 constituting the cohort (for logistical reasons, 20 patients were unable to attend the IgG4 test). A total of 344 individuals (35.4%) presented *L. loa* microfilaremia, 826 (85.1%) tested positive for the pan-IgG test, 771 (79.4%) for *L. loa* with the IgG4 test and 22 (2.3%) for onchocerciasis with the IgG4 test (Table 1).

The agreement in qualitative outcomes between both ARTs was estimated at 78.7% (764/971). In evaluating semi-quantitative results from both ARTs, the initial assessment revealed an overall agreement of 54.2% (526 out of 971 cases). This calculation assumed that a one-step error (a difference of one category between both ARTs, such as "1" vs. "2") was equivalent to a two ("1" vs. "3" or "0" vs. "2") or three-step error ("0" vs. "3"). To provide a more nuanced evaluation, errors were then weighted. The weighted analysis considers the severity of errors, giving higher importance to larger discrepancies between the results of the two methods. Specifically, a one-step error received a weight of 0.33, a two-step error was assigned a weight of 0.66, and a three-step error was given a weight of 1.00. With this weighted approach, the overall agreement increased significantly to 81.3% among the patients (Table 2). Kappa coefficients for qualitative and semi-quantitative results were estimated at 0.2744 (Standard deviation [SD]: 0.0315; $P < 0.0001$) and 0.1579 (SD: 0.0171; $P < 0.0001$, weighted results), respectively.

In the examination of the 207 discordant pairs (21.3%) between the pan-IgG test and the IgG4 test, 131 cases were individuals tested positive for loiasis by the pan-IgG test but negative by the IgG4 test. Among this group, 21 individuals were microfilaremic, 58 had exhibited at least once a loiasis-related sign, and 12 had both microfilaremia and a history of loiasis-related sign (with 24 cases having missing data). In contrast, 76 cases were found to be negative for loiasis on the pan-IgG test but positive on the IgG4 test. Within this group, 27 individuals had microfilaremia, 34 had exhibited at least once a loiasis-related sign, and 16 had both microfilaremia and a history of loiasis-related sign (with 11 cases having missing data). STH presence and elevated eosinophilia counts were less represented in the Positive pan-IgG test / Negative IgG4 test group (Table 3) than in the tested population. Female participants were more represented among the Negative pan-IgG test / Positive IgG4 test group than in tested population. Of the 22 subjects who were onchocerciasis-positive at the IgG4 test, 16 were also positive for loiasis on the IgG4 test and on the pan-IgG test, 4 were negative for loiasis on the IgG4 test but

**Table 1. Characteristics of the 971 participants tested with both pan-IgG and IgG4 tests.**

| Population characteristics | |
|---|---:|
| Sex-ratio (M/F) | 1.6 |
| Age (mean ± Standard deviation) (years) | 50.9 ± 14.8 |
| Presence of *Loa* mf | 344 (35.4%) |
| *L. loa* microfilarial density | |
| 0 mf/mL | 627 (64.6%) |
| 1–7999 mf/mL | 251 (25.9%) |
| 8000–19,999 mf/mL | 62 (6.4%) |
| ≥ 20,000 mf/mL | 31 (3.2%) |
| Pan-IgG test loiasis positivity | 826 (85.1%) |
| Score 1 | 787 (95.3%) |
| Score 2 | 34 (4.1%) |
| Score 3 | 5 (0.6%) |
| IgG4 test loiasis positivity | 771 (79.4%) |
| Score 1 | 529 (68.6%) |
| Score 2 | 167 (21.7%) |
| Score 3 | 75 (9.7%) |
| IgG4 test onchocerciasis positivity | 22 (2.3%) |
| Score 1 | 16 (72.7%) |
| Score 2 | 3 (13.6%) |
| Score 3 | 3 (13.6%) |
| Any STH infection* | 384 (39.5%) |
| Hookworm | 0 (0.0%) |
| *Ascaris lumbricoides* | 326 (33.6%) |
| *Trichuris trichiura* | 203 (20.9%) |

* Among the 971 individuals, 217 individuals (22.3%) did not have their stools examined for STH.

positive for loiasis on the pan-IgG test, 2 were negative for pan-IgG test but positive for loiasis on the IgG4 test and 0 were negative for loiasis for both ARTs. Of these 22 subjects, none had mfs in the skin.

Using pan-IgG test as the reference, overall sensitivity, specificity, area under the receiver operating characteristic (ROC) curve, positive and negative predictive values for loiasis result on the IgG4 test were estimated at 84.1% (95% confidence interval [95%CI]: 81.5–86.6%), 47.6% (95%CI: 39.2–56.0%), 0.659 (95%CI: 0.616–0.701), 90.1% (95%CI: 87.8–92.2%) and 34.5% (95%CI: 27.9–41.5%), respectively.

**Table 2. Qualitative and semiquantitative comparison of Pan-IgG test and IgG4 test results.**

| | | IgG4 test | | | | | |
|---|---|---|---|---|---|---|---|
| | | Score 0 | Score 1 | Score 2 | Score 3 | Positives (Scores 1–3) | Total (Scores 0–3) |
| Pan-IgG test | Score 0 | **69** | 58 | 16 | 2 | 76 | 145 |
| | Score 1 | 121 | **451** | 146 | 69 | 666 | 787 |
| | Score 2 | 9 | 19 | **4** | 2 | 25 | 34 |
| | Score 3 | 1 | 1 | 1 | **2** | 4 | 5 |
| | Positives (Scores 1–3) | 131 | 471 | 151 | 73 | **695** | 826 |
| | Total (Scores 0–3) | 200 | 529 | 167 | 75 | 771 | **971** |

**Table 3. Results from logistic regression models explaining discordances between ARTs.**

| | Positive pan-IgG test / Negative IgG4 test N = 746 | | | Negative pan-IgG test / Positive IgG4 test N = 591 | | |
|---|---|---|---|---|---|---|
| | aOR | 95% CI | p value | aOR | 95% CI | p value |
| **Age** (continuous) | 1.00 | 0.98–1.02 | 0.929 | 1.01 | 0.99–1.03 | 0.228 |
| **Sex** (ref. Female) | 0.73 | 0.46–1.16 | 0.184 | 0.47 | 0.27–0.84 | 0.010 |
| ***L. loa* microfilaremia** (continuous) | 1.00 | 1.00–1.00 | 0.910 | 1.00 | 1.00–1.00 | 0.346 |
| **Any STH presence** (ref. No) | 0.54 | 0.34–0.85 | 0.008 | 0.60 | 0.34–1.07 | 0.082 |
| **Calabar swelling** (ref. No) | 1.30 | 0.78–2.17 | 0.313 | 1.24 | 0.65–2.36 | 0.511 |
| **Eye worm** (ref. No) | 0.76 | 0.47–1.22 | 0.263 | 0.72 | 0.40–1.32 | 0.300 |
| **Eosinophilia count** (continuous) | 0.58 | 0.43–0.77 | <0.001 | 1.00 | 0.79–1.26 | 0.999 |
| **Pan-IgG test sq result of 2 or 3** (ref. 1) | 1.90 | 0.91–4.50 | 0.141 | N/A | | |
| **IgG4 test sq result of 2 or 3** (ref. 1) | N/A | | | 0.64 | 0.33–1.24 | 0.187 |

sq, semi-quantitative; N/A, not applicable

Table 4 presents diagnostic performance of each ART based on history or presence of loiasis-related signs. The missing data originate from the loiasis clinical questionnaires. Individuals were considered for these analyses only if both tests had been conducted and they had provided responses to the question pertaining to the reference used for each analysis.

Using the history of eye worm, or of Calabar swellings, or of both, as references (in the past year or lifetime), the specificity and sensitivity of the two ARTs were not significantly different (Table 4) with sensitivity ranging from 84.6 to 85.3% and specificity from 15.4 to 16.2% for the pan-IgG test, and sensitivity ranging from 78.1 to 80.7% and specificity from 20.3 to 21.9% for the IgG4 test.

Diagnostic performance, as assessed by the area under the ROC curve (AUC), was significantly better with the IgG4 test when the reference was defined as positive microfilaremia ($P = 0.0010$) and microfilaremia > 500 mf/mL ($P = 0.0423$). Diagnostic performance was significantly better with the pan-IgG test when reference was defined as occult loiasis ($P = 0.0374$)

## Discussion

In this study, we conducted a comprehensive evaluation of the IgG4-test, targeting both onchocerciasis (IgG4 antibodies to Ov-16 and OvOC3261) and loiasis (IgG4 to L1-SXP-1), marking the first such examination in a large population.

It is not possible to establish the sensitivity using as reference group all those who have been exposed to L3 infective larvae. Defining such a reference group would require a gold standard diagnostic for exposure, which does not exist. The current diagnostics only allow to rule in or out apparent clinical manifestations and/or the presence of microfilariae in the bloodstream, but do not say anything about exposure [7]. One can therefore only define a specificity compared to clinical manifestations and/or the presence of mfs.

Likewise, determining specificity would demand certainty that in the comparator group, no one has ever been exposed to infective L3, which again is not possible, as L3 do not necessarily mature into adult worms, mfs, and/or clinical signs. The specificities outlined in this paper should be interpreted as specificities-*like* i.e. as the likelihood of identifying symptoms (eye worm and/or Calabar swelling) and/or indicators of infection (microfilaremia). Indeed, specificity evaluation in the field is very challenging for antibodies due to uncertainties related to exposure. The reported specificities should, thus, be considered cautiously, recognizing its

**Table 4. Comparison of pan-IgG test and IgG4 test diagnostic performance.**

| | Prevalence | Pan-IgG test | | | | IgG4 test | | | | P-value[2] |
| | | Positive | Sensitivity [95% CI] | Specificity[1] [95% CI] | AUC [95% CI] | Positive | Sensitivity [95% CI] | Specificity[1] [95% CI] | AUC [95% CI] | |
|---|---|---|---|---|---|---|---|---|---|---|
| **Eye worm** | | | | | | | | | | |
| in the past year | 301/801 (37.6%) | 256/301 | 85.0% [80.5–88.9%] | 15.8% [12.7–19.3%] | 0.504 [0.478–0.530] | 243/301 | 80.7% [75.8–85.9%] | 21.6% [18.1–25.5%] | 0.512 [0.483–0.540] | 0.6903 |
| in the lifetime | 387/802 (48.3%) | 330/387 | 85.3% [81.3–88.6%] | 16.1% [12.7–20.0%] | 0.507 [0.482–0.532] | 312/387 | 80.6% [76.3–84.4%] | 21.9% [18.0–26.2%] | 0.513 [0.485–0.541] | 0.7784 |
| **Calabar swelling** | | | | | | | | | | |
| in the past year | 202/800 (25.3%) | 171/202 | 84.7% [78.9–89.3%] | 15.4% [12.6–18.5%] | 0.500 [0.471–0.529] | 160/202 | 79.2% [73.0–84.6%] | 20.7% [17.6–24.2%] | 0.500 [0.467–0.532] | 0.9802 |
| in the lifetime | 247/800 (30.9%) | 209/247 | 84.6% [79.5–88.9%] | 15.4% [12.5–18.7%] | 0.500 [0.473–0.527] | 193/247 | 78.1% [72.5–83.1%] | 20.3% [17.0–23.8%] | 0.492 [0.461–0.523] | 0.6654 |
| **Eye worm and/or Calabar swelling** | | | | | | | | | | |
| in the past year | 388/802 (48.4%) | 331/388 | 85.3% [81.4–88.7%] | 16.2% [12.8–20.1%] | 0.507 [0.482–0.532] | 309/388 | 79.6% [75.3–83.5%] | 21.0% [17.2–25.3%] | 0.503 [0.475–0.531] | 0.7658 |
| in the lifetime | 451/803 (56.0%) | 384/451 | 85.1% [81.5–88.3%] | 16.2% [12.5–20.5%] | 0.507 [0.481–0.532] | 360/451 | 79.8% [75.8–83.4%] | 21.2% [17.1–26.0%] | 0.506 [0.477–0.534] | 0.9154 |
| **Microfilaremia** | | | | | | | | | | |
| Any level of mf/mL | 334/971 (34.4%) | 292/334 | 87.4% [82.4–91.2%] | 15.7% [13.1–18.5%] | 0.515 [0.405–0.539] | 296/334 | 88.6% [84.7–91.8%] | 25.4% [22.1–29.0%] | 0.570 [0.546–0.594] | 0.0010 |
| > 500 mf/mL | 243/971 (25.0%) | 212/243 | 87.3% [82.5–91.2%] | 15.7% [13.1–18.5%] | 0.515 [0.490–0.540] | 211/243 | 86.8% [81.9–90.8%] | 23.1% [20.1–26.3%] | 0.550 [0.523–0.576] | 0.0423 |
| > 2,500 mf/mL | 166/971 (17.1%) | 141/166 | 84.9% [78.6–90.0%] | 14.9% [12.5–17.6%] | 0.499 [0.469–0.529] | 136/166 | 81.9% [75.2–87.5%] | 21.1% [18.4–24.1%] | 0.515 [0.483–0.548] | 0.4033 |
| >10,000 mf/mL | 74/971 (7.6%) | 62/74 | 83.8% [73.4–91.3%] | 14.8% [12.6–17.3%] | 0.493 [0.449–0.537] | 59/74 | 79.7% [68.8–88.2%] | 20.6% [18.0–23.4%] | 0.502 [0.454–0.550] | 0.7209 |
| **Any sign of *Loa loa* presence[3]** | 613/853 (71.9%) | 530/613 | 86.5% [83.5–89.1%] | 17.9% [13.3–23.4%] | 0.522 [0.494–0.550] | 506/613 | 82.5% [79.3–85.5%] | 26.3% [20.8–32.3%] | 0.544 [0.512–0.576] | 0.3039 |
| **Occult loiasis[4]** | 279/821 (34.7%) | 238/279 | 85.3% [80.6–89.2%] | 15.8% [12.8–19.3%] | 0.506 [0.480–0.532] | 210/279 | 75.3% [69.8–80.2%] | 18.5% [15.3–22.1%] | 0.469 [0.439–0.499] | 0.0374 |

[1] Specificity refers to the probability of each test to "detect" the symptom (eyeworm, Calabar swelling. . .)–Specificity defined as the probability to detect loiasis and only loiasis should be evaluated in the lab

[2] Comparison of Area under the ROC curve (AUC) values

[3] Eyeworm in the lifetime and/or Calabar swelling in the lifetime and/or microfilaremia

[4] Eyeworm in the lifetime and/or Calabar swelling in the lifetime and no microfilaremia

limitations in reflecting the true specificity compared to other pathogens. The specificity *per se*, indicating the probability of false positives due to other infections, needs to be evaluated in a laboratory setting using well characterized sera from non-*Loa* endemic areas. This crucial step is imperative for the comprehensive assessment of the new IgG4-test prototype.

We created the variable 'Any sign of *Loa loa* presence', yielding an overall prevalence of loiasis at 71.9%. It is recognized that diagnostic performance varies with the prevalence of infection in a given population [14]. Reevaluation of these diagnostic performance in a hypoendemic context for loiasis could provide valuable insights.

The pan-IgG test exhibited a sensitivity of 87.4%, while the prototype IgG4 test demonstrated a slightly higher sensitivity at 88.6% to detect any level of microfilaremia. When utilizing eye worm or Calabar swelling as references, both rapid tests showed similar sensitivity. However, the diagnostic performance varied when microfilaremia and occult loiasis were used as references. This unexpected result warrants further investigation. IgG4 has been identified as the predominant IgG isotype in *L. loa* infection. Interestingly, anti-*Loa* IgG4 recognizes not only antigens from adult worms but also those from L3 and microfilarial stages. Notably, there is no correlation between microfilarial densities and elevated levels of anti-*Loa* IgG4 [15,16].

To identify high microfilarial densities (> 10,000 mf/mL), the pan-IgG test demonstrates higher sensitivity compared to the IgG4 test. Similarly, when focusing on occult loiasis, the pan-IgG test also shows higher sensitivity than the IgG4 test. This could be due to the presence of another IgG isotype, necessitating further investigations.

We have also shown that diagnostic performance of both ARTs decrease as the microfilaremia of individuals increases. Veletzky *et al.* also found that individuals with high MFD often tested negative by the pan-IgG ART and ELISA [17]. Several hypotheses can be proposed: (i) the overproduction of mfs could block the antibody reaction mechanically or by direct binding with the antibody, (ii) high microfilaremia may be due to lower levels of immunity, resulting in reduced antibody levels, or (iii) the overproduction of microfilariae could lead to alterations in the spleen, as described in various articles[18–21], causing a reduction in splenic function and, consequently, altered antibody production. These hypotheses require further study.

No individual showed *O. volvulus* mfs in the skin, making it difficult to assess positive cross-reactivity between onchocerciasis and loiasis results from the IgG4 test. Further studies in co-endemic areas for other filariasis (onchocerciasis, *M. perstans* or lymphatic filariasis) are needed to explore this aspect.

The IgG4 test displayed a sensitivity of 84.1% and a specificity of 47.6% for detecting loiasis compared to the pan-IgG test. This led to the estimation of a Kappa coefficient at $0.2744 \pm 0.0315$, indicating the level of agreement in the qualitative results between the two ARTs. While the estimated sensitivity is overall good and suggests the potential utility of a biplex as a tool to map loiasis endemic areas, the specificity seems very poor but need to be reevaluated in the laboratory using sera.

Analysis of the 207 discordant pairs revealed different patterns. Among these pairs, 131 showed positive results in the pan-IgG test but negative in the IgG4-test, while 76 exhibited the opposite pattern. Microfilaremia was present in 19.6% of cases in the former group and 41.5% in the latter. Notably, the group with positive pan-IgG test and negative IgG4-test results showed a reduced representation of STH presence and of elevated eosinophilia counts compared to the overall tested population. Since the IgG4 test is designed to be more specific, the pan-IgG test positivity might be a false positive. This could be due to the presence of soil-transmitted helminths (STH) or another parasitic infection causing high eosinophilia. Conversely, the group with negative pan-IgG test and positive IgG4-test results exhibited a higher proportion of female participants than the overall population. This result remains to be explained.

The logistic regression models indicated that the semi-quantitative intensity of the ARTs and microfilaremia were not associated with the likelihood of having discordant test results. Additionally, the specificity and sensitivity of the two ARTs did not significantly differ when using the history of eye worm, Calabar swellings, or both as references. However, the diagnostic performance was notably better with the IgG4-test when positive microfilaremia was the

reference, while the pan-IgG test outperformed when occult loiasis was considered. The presence of several discordant results prompts further consideration. The divergence in analytes being analyzed may contribute to these discrepancies, necessitating thoughtful exploration to unravel the underlying factors. While various strategies have been evaluated to combat onchocerciasis in areas where it is hypo-endemic and co-endemic areas for loiasis, none have been retained yet [22]. This underscores the need for alternative diagnostic tools that can identify individuals at risk of serious side effects. Although IgG4-test do not fully address this issue, it may contribute to the diagnosis and mapping of loiasis. This can help identify areas at risk and offer a cost-effective solution for managing both onchocerciasis and loiasis.

## Acknowledgments

We thank the French Embassy in Republic of Congo. We thank the Lékoumou health district, the medical, paramedical and technical staff of the Sibiti hospital, the PNLO and IRD drivers, and the participants for agreeing to participate.

## Author Contributions

**Conceptualization:** Jérémy T. Campillo, Marco A. Biamonte.

**Data curation:** Jérémy T. Campillo.

**Formal analysis:** Jérémy T. Campillo.

**Funding acquisition:** Cédric B. Chesnais.

**Investigation:** Jérémy T. Campillo, Marlhand C. Hemilembolo, François Missamou, Cédric B. Chesnais.

**Methodology:** Jérémy T. Campillo.

**Validation:** Marco A. Biamonte.

**Writing – original draft:** Jérémy T. Campillo.

**Writing – review & editing:** Jérémy T. Campillo, Marco A. Biamonte, Marlhand C. Hemilembolo, François Missamou, Michel Boussinesq, Sébastien D. S. Pion, Cédric B. Chesnais.

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
