## [Decision Letter · Decision Letter 0]

19 Aug 2024

Dear Dr Campillo,

Thank you very much for submitting your manuscript "Evaluation of a novel biplex rapid diagnostic test for antibody responses to Loa loa and Onchocerca volvulus infections" for consideration at PLOS Neglected Tropical Diseases. As with all papers reviewed by the journal, your manuscript was reviewed by members of the editorial board and by several independent reviewers. In light of the reviews (below this email), we would like to invite the resubmission of a significantly-revised version that takes into account the reviewers' comments. 

Thanks you for submitting your manuscript, your manuscript has been evaluated as an acceptable one after some revisions. You are kindly requested to correct all the amendment requested by the reviewer. Also, make sure the revised manuscript follows our publication guidelines.

We cannot make any decision about publication until we have seen the revised manuscript and your response to the reviewers' comments. Your revised manuscript is also likely to be sent to reviewers for further evaluation.

Sincerely,

Jong-Yil Chai

Section Editor

Jong-Yil Chai

Section Editor

Thanks you for submitting your manuscript, your manuscript has been evaluated as an acceptable one after some revisions. You are kindly requested to correct all the amendment requested by the reviewer. Also, make sure the revised manuscript follows our publication guidelines.

Reviewer's Responses to Questions

**Key Review Criteria Required for Acceptance?**

**Methods**

-Are the objectives of the study clearly articulated with a clear testable hypothesis stated?

-Is the study design appropriate to address the stated objectives?

-Is the population clearly described and appropriate for the hypothesis being tested?

-Is the sample size sufficient to ensure adequate power to address the hypothesis being tested?

-Were correct statistical analysis used to support conclusions?

-Are there concerns about ethical or regulatory requirements being met?

Reviewer #1: the objectives of the study are clearly articulated with a clear testable hypothesis stated

the study design is appropriate to address the stated objectives

the population is clearly described and appropriate for the hypothesis being tested

the sample size is sufficient to ensure adequate power to address the hypothesis being tested

statistical analysis used were correct to support conclusions

there are no concerns about ethical or regulatory requirements being met

Reviewer #2: -Are the objectives of the study clearly articulated with a clear testable hypothesis stated?

Yes

-Is the study design appropriate to address the stated objectives?

Yes

-Is the population clearly described and appropriate for the hypothesis being tested?

Yes

-Is the sample size sufficient to ensure adequate power to address the hypothesis being tested?

Yes

-Were correct statistical analysis used to support conclusions?

Yes

-Are there concerns about ethical or regulatory requirements being met?

No

Reviewer #3: This paper contributes to the evaluation of diagnostic tests for onchocerciasis and loasis which are important concerns for the onchocerciasis elimination program.

This publication meets the essential criteria for acceptance.

**Results**

-Does the analysis presented match the analysis plan?

-Are the results clearly and completely presented?

-Are the figures (Tables, Images) of sufficient quality for clarity?

Reviewer #1: the analysis presented match the analysis plan

the results are clearly and completely presented

the figures (Tables, Images) are of sufficient quality for clarity

Reviewer #2: -Does the analysis presented match the analysis plan?

Yes

-Are the results clearly and completely presented?

Yes

-Are the figures (Tables, Images) of sufficient quality for clarity?

Reviewer #3: The results are well presented, clear and match with the analysis plan

**Conclusions**

-Are the conclusions supported by the data presented?

-Are the limitations of analysis clearly described?

-Do the authors discuss how these data can be helpful to advance our understanding of the topic under study?

-Is public health relevance addressed?

Reviewer #1: the conclusions are supported by the data presented

the limitations of analysis are clearly described

the authors discuss how these data can be helpful to advance our understanding of the topic under study

public health relevance is addressed

Reviewer #2: -Are the conclusions supported by the data presented?

Yes, however I think some of the findings and implications of them should be discussed more clearly.

-Are the limitations of analysis clearly described?

The limitations of the presented data should be discussed more clearly.

-Do the authors discuss how these data can be helpful to advance our understanding of the topic under study?

Yes, but I suggest that the authors should revise and improve the discussion. There are various interesting findings:

e.g.

In the author summary they state “…its performance varied according to the density of microfilariae in the blood.” And this is also visible in table 4. And how do the authors explain the differences in test performance depending on reference diagnostics?

I suggest to better discuss this finding and compare it to available data.

In the abstract conclusion the authors state that “…IgG and IgG4 antibody responses should be interpreted differently” 

This is an interesting finding, but it is not further discussed. What could be the implications? 

-Is public health relevance addressed?

I suggest to further elaborate this point in the discussion – do the author’s findings provide new possibilities for loiasis diagnostics or mapping strategies? If not, why? What could be improved in the development of future tests?

Reviewer #3: No onchocerciasis patient was formally identified, further studies were suggested, and this should be done in onchocerciasis known endemic areas.

**Editorial and Data Presentation Modifications?**

Reviewer #1: no specific comment

Reviewer #2: Line 34: “Diagnosing loiasis is challenging due to sporadic and non-specific clinical symptoms.”

- This sentence may be misread, rephrase for clarity.

Line 35 and later: I would call them larvae and not embryos 

Line 49: do you mean “experienced”?

Line 63: do you mean “underlines”?

Line 67: do you mean swelling<s> or maybe add more details? And I suggest to rephrase “Treating it is tricky,…”

Line 68: state the type of test that you tested (i.e. rapid diagnostic…)

Line 82: 40% seems quite high especially if you are referring to the total population (and not only adults).

Line 85-86: add citations for symptoms, the provided citations do not describe these general symptoms. 

Lines 94-96: Is it only (host) genetics or is the situation more complex?

Line 104-106: rephrase for clarity

Lines 112-113: I think “of antibodies” may be removed from the sentence and maybe: ll-SXP-1, a marker of exposure to loa loa infection in the laboratory setting or similar.

Line 119: remove the second “as possible” and lines 119-123: this phrase is too long, rephrase for clarity.

Line 167: why were the slides dehemoglobinized before staining?

Line 169: how were discordant results handled?

Line 183: howe were eyeworm and calabar swelling queried? Using the raploa questionnaire or similar methods?

Line 201: I think it should say “infection was assessed”

 Lines 233-230: As I understand the presentations, the numbers don’t add up. Maybe redefine for clarity.

Line 242-243: in “the” tested population

Line 269: for “the” IgG test…

Lines 330-332: rephrase for clarity

Line 324: sera?

Reviewer #3: test-and-(not)-treat should be well noted Test and not Treat (TaNT)

**Summary and General Comments**

Reviewer #1: This a very interesting manuscript on a new diagnostic test for field application on a difficult disease. The text is well writtten and reads well. 

There are however some very minor comments .

Page 7, line 46: “discordance […] were investigated” please add a “s” at discordance.

Page 17, line 223-224: some definition of the one step error (or more) would be helpful to the reader that is not familiar with validation processes

Page 19, table 3: number of participants(N=) for each discordance model could be added for transparency

Page 22, lines 290-295: wouldn’t it be possible to define a reference standard with IgG testing complemented with clinical manifestations and compare the IgG4 based test to get a more true sensitivity and specificity?

Reviewer #2: This paper by Campillo et al describes a comparison of two different rapid diagnostic tests detecting IgG and IgG4 antibodies against Loa loa. Diagnosis of loiasis remains difficult and data on RDT performance, especially from endemic populations, is limited. 

The authors clearly describe the used methods and present the data straight forward. However, I think that certain points deserve more attention, should be put into context and the discussion should be expanded (see points below).

Reviewer #3: This paper contributes to the evaluation of diagnostic tests for onchocerciasis and loasis, which are major concerns for the onchocerciasis elimination program. The major concern for the Test and Not Treat (TaNT) strategy is the identification of subjects at risk of Serious Adverse Events (SAEs). The LoaScope exists and is used for this strategy in onchocerciasis hypo-endemic areas and co-endemic areas for loasis. However, there are constraints associated with this tool. It would therefore be useful to work on other diagnostic tools that can be used to identify subjects at risk of serious side-effects. The diagnostic tools presented in this paper do not solve this problem, but at least contribute to the diagnosis of loasis, which can contribute to the mapping of loasis and thus help to identify areas at risk. The fact that these tools can be used for both onchocerciasis and loasis presents a major advantage in terms of cost.

PLOS authors have the option to publish the peer review history of their article (what does this mean?). If published, this will include your full peer review and any attached files.

Reviewer #1: No

Reviewer #2: No

Reviewer #3: No
---

## [Editor Report · Decision Letter 1]

23 Sep 2024

Dear , Salvador,

We are pleased to inform you that your manuscript 'Evaluation of a novel biplex rapid diagnostic test for antibody responses to Loa loa and Onchocerca volvulus infections' has been provisionally accepted for publication in PLOS Neglected Tropical Diseases.

Best regards,

Dawit Gebeyehu Getachew, MPH

Guest Editor

Jong-Yil Chai

Section Editor

---

## [Editor Report · Acceptance letter]

7 Oct 2024

Dear Dr Campillo,

We are delighted to inform you that your manuscript, "Evaluation of a novel biplex rapid diagnostic test for antibody responses to Loa loa and Onchocerca volvulus infections," has been formally accepted for publication in PLOS Neglected Tropical Diseases.

Best regards,

Shaden Kamhawi

co-Editor-in-Chief

Paul Brindley

co-Editor-in-Chief
